# Graph Convolutional Network and Contrastive Learning Small Nucleolar RNA (snoRNA) Disease Associations (GCLSDA): Predicting snoRNA–Disease Associations via Graph Convolutional Network and Contrastive Learning

**DOI:** 10.3390/ijms241914429

**Published:** 2023-09-22

**Authors:** Liangliang Zhang, Ming Chen, Xiaowen Hu, Lei Deng

**Affiliations:** School of Computer Science and Engineering, Central South University, Changsha 410083, China; 8202200114@csu.edu.cn (L.Z.); 8209200609@csu.edu.cn (M.C.); altriavin@csu.edu.cn (X.H.)

**Keywords:** *snoRNA*–disease association, graph convolutional network, contrastive learning, bipartite graph

## Abstract

Small nucleolar RNAs (snoRNAs) constitute a prevalent class of noncoding RNAs localized within the nucleoli of eukaryotic cells. Their involvement in diverse diseases underscores the significance of forecasting associations between snoRNAs and diseases. However, conventional experimental techniques for such predictions suffer limitations in scalability, protracted timelines, and suboptimal success rates. Consequently, efficient computational methodologies are imperative to realize the accurate predictions of snoRNA–disease associations. Herein, we introduce GCLSDA—**g**raph Convolutional Network and **c**ontrastive **l**earning predict **s**noRNA **d**isease **a**ssociations. GCLSDA is an innovative framework that combines graph convolution networks and self-supervised learning for snoRNA–disease association prediction. Leveraging the repository of MNDR v4.0 and ncRPheno databases, we construct a robust snoRNA–disease association dataset, which serves as the foundation to create bipartite graphs. The computational prowess of the light graph convolutional network (LightGCN) is harnessed to acquire nuanced embedded representations of both snoRNAs and diseases. With careful consideration, GCLSDA intelligently incorporates contrast learning to address the challenging issues of sparsity and over-smoothing inside correlation matrices. This combination not only ensures the precision of predictions but also amplifies the model’s robustness. Moreover, we introduce the augmentation technique of random noise to refine the embedded snoRNA representations, consequently enhancing the precision of predictions. Within the domain of contrast learning, we unite the tasks of contrast and recommendation. This harmonization streamlines the cross-layer contrast process, simplifying the information propagation and concurrently curtailing computational complexity. In the area of snoRNA–disease associations, GCLSDA constantly shows its promising capacity for prediction through extensive research. This success not only contributes valuable insights into the functional roles of snoRNAs in disease etiology, but also plays an instrumental role in identifying potential drug targets and catalyzing innovative treatment modalities.

## 1. Introduction

The investigation into small nuclear RNAs (snoRNAs) and their correlation with diseases has gained substantial momentum in recent years. snoRNAs, a class of small noncoding RNAs, are widely distributed within the nucleus of eukaryotic cells [1]. In vertebrates, the genes encoding snoRNAs predominantly inhabit the intronic regions of both protein-coding and non-protein-coding genes, undergoing subsequent post-transcriptional processing to culminate in mature snoRNAs [2]. These snoRNAs are characterized by conserved structural motifs. In earlier studies, snoRNAs were broadly categorized into C/D box snoRNAs and H/ACA box snoRNAs [3], predominantly associated with the modulation of $2^{\prime }$-O-ribose methylation and pseudouridylation of ribosomal RNA. Nevertheless, recent decades have unveiled a growing subset of snoRNAs devoid of specific targets or well defined cellular functions, hinting at the untapped potential of snoRNAs in disease control [4]. Their active involvement in nucleoside modifications emphasizes their role in various human diseases. Notably, host genes harboring snoRNA coding sequences within their introns exhibit differential expression patterns in disease contexts [5]. The perturbation of the snoRNA expression or function correlates with diverse ailments, encompassing cancers, neurological disorders, immune conditions, and genetic anomalies [4,5,6].

Deciphering the intricate interplay between snoRNAs and diseases holds immense potential for groundbreaking advancements in diagnostic modalities and therapeutic strategies. For instance, snoRNAs may emerge as promising biomarkers for early disease detection and promising targets for innovative therapeutic interventions. Profound shifts in snoRNA expression within tumor cells, tissues, and bodily fluids could yield novel biomarkers for cancer diagnosis and therapeutic exploration. Furthermore, delving into the interplay between snoRNAs and diseases can furnish invaluable insights into the fundamental mechanisms underpinning disease onset and progression. By discerning the role of snoRNAs in healthy cellular contexts and deciphering their dysregulation’s contributions to disease states, researchers can identify novel avenues and targets for therapeutic intervention, thereby enriching our arsenal against various ailments.

Currently, prediction methods for RNA–disease associations can be broadly classified into three categories: machine learning-based, network analysis-based, and deep learning-based approaches. Machine learning-based methods: This category encompasses techniques that utilize known snoRNA–disease association data to construct prediction models through various machine learning algorithms, enabling the prediction of novel associations. For instance, PSnoD [7] proposes a method based on bounded kernel norm regularization to predict the association between snoRNA and disease. This method constructs an association matrix between snoRNA and disease, in which each element represents the similarity or degree of association between snoRNA and disease. SnoDi-LSGT [8] proposes a method based on local similarity constraints and global topology constraints to predict the association between snoRNA and diseases. This method combines local similarities and global topological features of snoRNAs and diseases to improve prediction accuracy and interpretability. Similarly, iSnoDi-MDRF [9] proposes a method based on multiple biological data and a ranking framework to predict the association between snoRNA and disease. By integrating multiple biological data such as gene expression data, gene function annotations, protein interaction networks, etc., and using a ranking framework to rank features.

Network analysis-based approach: This approach leverages bioinformatics and network analysis tools to investigate the functional interplay between RNA and diseases, often employing co-expression networks and protein interaction networks. For example, Sun et al. [10] proposed a global network-based calculation method RWRlncD based on lncRNA functional similarity network. In the study, after successively constructing the lncRNA-disease association network, disease similarity network and lncRNA functional similarity network, RWRlncD predicted potential lncRNA-disease relationships by performing random walk restart (RWR) on the lncRNA functional similarity network. In addition to single-layer networks, some researchers have tried to construct lncRNA-disease multi-level networks, and based on this multi-level network to study and identify new disease-related lncRNAs. Yao et al. [11] proposed the algorithm LncPriCNet based on a multi-layer composite network to predict disease-related lncRNA. The study constructed a composite network by combining phenotype-phenotype interactions, lncRNA-lncRNA interactions, and gene-gene interactions with disease-ncRNA relationships, and then used the random walk restart algorithm (RWR) to predict related lncRNAs.

Deep learning-based approach: This class of methods harnesses the power of deep learning algorithms to acquire the embedded representations of RNA and diseases for predicting new associations. For example, GANCDA [12] proposes a method based on deep generative adversarial networks to predict the association between circRNA and diseases. This method learns the potential association between circRNA and disease by converting circRNA and disease data into feature representations and conducting adversarial training through a generator network and a discriminator network.

The aforementioned prediction models offer various perspectives and avenues for researching the relationship between snoRNAs and diseases, providing informative references for understanding disease mechanisms and snoRNA functions. However, these methods encounter certain challenges. Machine learning-based approaches may face difficulties in feature selection, overfitting, and adapting to complex datasets. Network analysis-based methods might not adequately account for higher-order associations within intricate networks. Deep learning-based techniques could encounter issues related to overfitting and limited interpretability. Consequently, there is a need to refine these methods to enhance the prediction accuracy and interpretability.

In recent years, contrastive learning and graph neural networks have gained significant traction across various research domains. The former entails learning data representations by juxtaposing the similarities and differences between samples, while the latter effectively models interactions between nodes and edges in graphs, enabling both representation learning and prediction tasks. However, certain challenges persist within these methods. Notably, perturbations in the graph structure can lead to the loss of crucial information, thereby causing deviations in the results. To address these issues, we introduce a novel framework named GCLSDA in this study. In our previous study from 2021, we introduced GCNSDA [13], a pioneering framework utilizing graph convolutional networks for predicting associations between snoRNAs and diseases. This work marked a crucial milestone by showcasing the potential of computational methodologies to address the challenges of snoRNA–disease association predictions. Building upon this foundation, our latest research presents GCLSDA, an advanced framework that represents the natural evolution of our efforts. While GCNSDA laid the groundwork for employing GCNs in this context, GCLSDA extends this approach by integrating contrastive learning, random noise augmentation, and harmonizing contrast and recommendation tasks. These enhancements substantially bolster prediction precision and model robustness, further advancing our quest to unlock the potential of computational methodologies in predicting snoRNA–disease associations.

GCLSDA adopts a self-supervised approach for predicting snoRNA–disease associations, harnessing the capabilities of the light graph convolutional network (LightGCN) [14]. LightGCN is a renowned graph-based recommendation model celebrated for its simplicity and effectiveness within recommendation systems. Its primary goal is to address the scaling and model complexity issues that collaborative filtering-based recommendation frequently faces. Neighborhood aggregation, a key method used in graph convolution networks, is at the core of LightGCN’s success in producing reliable suggestions. The central objective of the GCLSDA framework is to overcome the challenges posed by data sparsity and noise within correlation matrices. This is achieved through the incorporation of contrast learning, which not only enhances prediction accuracy but also fortifies model robustness. For the construction of a dependable snoRNA–disease association dataset, we leverage two key databases, namely MNDR v4.0 [15] and ncRPheno [16]. The LightGCN is then harnessed to acquire embedded representations of both snoRNAs and diseases. In a bid to enhance the dataset, we introduce random noise to the embedded snoRNA representations, deviating from conventional edge and node-dropping techniques. Within the realm of contrast learning, we unify the contrast and recommendation tasks, employing consistent noise perturbations for cross-layer contrast. This strategic approach simplifies the propagation mechanism and concurrently reduces computational complexity.

## 2. Results and Discussions

In this section, we outline the outcomes of our experimental endeavors utilizing the GCLSDA method for the prediction of snoRNA–disease associations.

### 2.1. Experimental Setup

To rigorously assess the predictive prowess of our experimental methodology, we employed a systematic and unbiased approach by conducting performance evaluations using both five-fold cross-validation and ten-fold cross-validation techniques [17]. These methods allowed us to comprehensively gauge the predictive capability of the GCLSDA model.

The dataset under consideration contains verified or established associations, which we randomly divided into k distinct subsets. For each iteration of cross-validation, one of these subsets was designated as the test set, while the remaining k-1 subsets were amalgamated to compose the training set. In order to ensure equitable evaluation conditions, we generated an equal number of negative samples at random for the test set. Notably, during the utilization of the training set, the negative sample generation process was omitted, as the algorithm itself encompasses this procedure.

Regarding the potential presence of latent associations within the negative samples, their impact is deemed negligible due to their limited proportion within the overall unverified sample pool. Consequently, their influence on the experimental outcomes can be safely disregarded.

### 2.2. Evaluation Criteria

To assess the predictive ability of our model in a more intuitive and understandable manner, we utilize widely adopted metrics in the industry, including the area under the curve (AUC) and area under the precision–recall curve (AUPR). These metrics provide a comprehensive measure of the model’s performance.

The calculation formulas for the evaluation metrics are as follows:(1)$$\begin{array}{cc} FPR & =\frac{FP}{FP+TN} \end{array}$$(2)$$\begin{array}{cc} TPR & =\frac{TP}{TP+FN} \end{array}$$(3)$$\begin{array}{cc} Recall & =\frac{TP}{TP+FN} \end{array}$$(4)$$\begin{array}{cc} Precision & =\frac{TP}{TP+FP} \end{array}$$

*TP* and *FP* represent the correct and incorrect classification results on the existing snoRNA–disease adversarial samples, respectively. *TN* and *FN* represent the correct and incorrect classification results on unrelated sample sets, respectively. The true positive rate (*TPR*), representing the proportion of correctly identified positive cases, and the false positive rate (*FPR*), indicating the proportion of incorrectly identified negative cases, are calculated. By adjusting the decision threshold, we can construct the ROC curve [18] and PR curve [19], with *FPR* as the *x* axis and *TPR* as the *y* axis. The AUC and AUPR values are then calculated from these curves, providing quantitative measures of the model’s accuracy. A higher AUC value indicates a better prediction performance of the model.

### 2.3. Performance Evaluation for GCLSDA

To ensure the robustness and reliability of our performance evaluation and to mitigate the influence of random fluctuations, we employed both five-fold cross-validation and ten-fold cross-validation methodologies to comprehensively appraise the predictive performance of the GCLSDA model in the domain of snoRNA–disease association prediction. The results of the 5-fold cross-validation are summarized in Table 1, encompassing the computed AUC and AUPR values. Additionally, the visual representations of the corresponding receiver operating characteristic (ROC) and precision–recall (PR) curves are depicted in Figure 1A,B, respectively. Correspondingly, the outcomes of the ten-fold cross-validation are provided in Table 2, while Figure 1C,D portray the associated ROC and PR curves.

By conducting cross-validation and subsequently assessing the AUC and AUPR metrics, we can effectively gauge the performance of the model in predicting snoRNA–disease associations. These evaluation metrics offer valuable insights into the model’s predictive prowess, shedding light on its capacity to reliably discriminate between positive and negative samples and affording a comprehensive overview of its overall predictive performance.

### 2.4. Comparison with Other Models

In the context of snoRNA–disease association prediction, given its relative novelty, there exists a dearth of established prediction methodologies. Consequently, we turned to analogous prediction methods from related biological domains—namely lncRNA–disease, circRNA–disease, and miRNA–disease associations—as a benchmark for assessing the efficacy of our proposed GCLSDA model.

In this comparative analysis, we juxtaposed our GCLSDA model against six closely related methods: KATZHMDA [20], DMFMDA [21], NTSHMDA [22], SDLDA [23], DMFCDA [24], and GCNMDA [25]. KATZHMDA operates on the KATZ algorithm for predicting microbe–disease associations, whereas DMFMDA leverages neural networks to predict these associations after mapping one-hot encoded microbe and disease IDs to low-dimensional vectors. SDLDA combines the SVD algorithm with a deep neural network for lncRNA–disease association prediction, while DMFCDA employs deep neural networks to derive nonlinear features from circRNA–disease association matrices. GCNMDA leverages graph neural networks and conditional random fields for drug-disease association inference, while NTSHMDA employs a random walk with a restart-based approach to predict microbe–disease associations.

**Figure 1 ijms-24-14429-f001:**
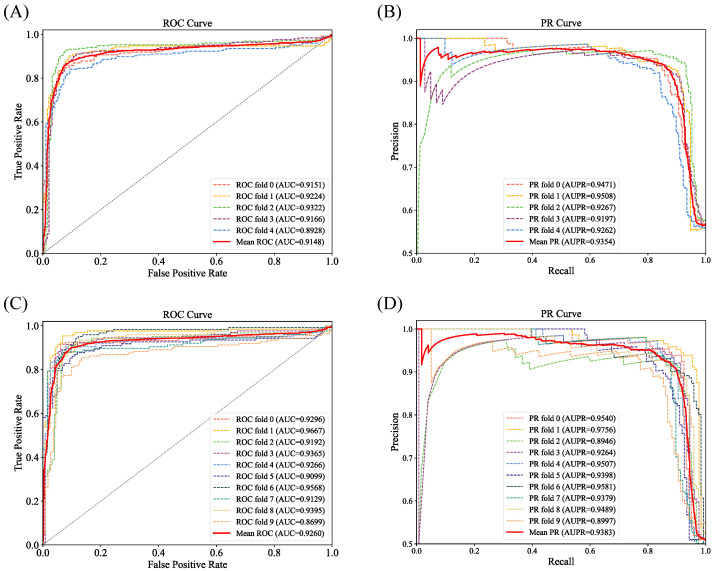
ROC and PR curves yielded by GCLSDA in five-fold and ten-fold. (**A**) ROC curve in five-fold. (**B**) PR curve in five-fold. (**C**) ROC curve in ten-fold. (**D**) PR curve in ten-fold.

The results of our experiments in Table 3, conducted using five-fold cross-validation, revealed that GCLSDA exhibited superior performance, achieving an average AUC of 0.9148 and an average AUPR of 0.9354. Notably, these metrics outperformed the second-best method, GCNMDA, by 4.87% and 6.28%, respectively.

For the sake of enhanced visual clarity in facilitating comparisons, we presented the corresponding ROC curves in Figure 2. The discernible trend highlighted by these results is the efficacy of GCLSDA in prognosticating novel snoRNA–disease associations. This robust performance is underpinned by the fundamental disparity in methodology between GCLSDA and traditional approaches. The conventional methods often generate similar features for both snoRNAs and diseases, inadvertently incorporating noise attributed to the scarcity of biological data. This noise, in turn, hampers prediction accuracy. In stark contrast, our model circumvents the need for similarity calculations and instead capitalizes on the learning capabilities of graph neural networks. This approach fully accounts for the influence of neighboring nodes within the bipartite graph encompassing snoRNAs and diseases. Additionally, it explicitly integrates the impact of high-order neighbor features on the present node within the network. This innovative approach substantially contributes to GCLSDA’s superior predictive performance.

### 2.5. Ablation Study

In the context of the ablation study conducted on GCLSDA, we initiated our investigation by performing an experiment involving the LightGCN. This particular endeavor sought to independently evaluate the predictive capabilities of LightGCN in the context of snoRNA–disease association prediction, unassisted by the comprehensive enhancements provided by the GCLSDA framework.

Subsequently, our attention shifted towards a meticulous examination of the individual contributions rendered by the self-supervised graph learning (SGL) graph enhancement technique. To achieve this, we executed two distinct ablation experiments on the selected SGL model, aptly termed SGL with edge-dropout (SGL(ED)) and SGL with node-dropout (SGL(ND)).

In the SGL(ED) ablation experiment, an intentional removal of edges from the SGL model was executed, performed with a specified probability. The objective was to ascertain the influence of this edge-removal strategy on the prediction of snoRNA–disease associations. Conversely, the SGL(ND) ablation experiment entailed the elimination of specific nodes alongside the edges connected to those nodes. This elimination process was executed with a particular probability, enabling a comprehensive assessment of how the removal of nodes and their associated edges impacted the predictive performance.

A carefully thought-out five-fold cross-validation methodology was used to objectively evaluate each ablation experiment. The results of these ablation experiments were compared to those of the thorough GCLSDA model, and this allowed us to gain an important understanding of the substantive relevance and unique contributions included inside the SGL graph enhancement method. Additionally, this enabled us to draw broad conclusions about the GCLSDA framework’s general effectiveness in terms of predicting snoRNA–disease connections. The intricate details of these experimental outcomes are diligently presented in Table 4. According to the experimental results, we observe that the comprehensive GCLSDA model outperformed the individual components, attaining an AUC of 0.9148 and an AUPR of 0.9354. These results highlight the effectiveness of the SGL enhancements and the synergistic benefits they bring when integrated into the GCLSDA framework.

#### Effects of Parameters

During our experimentation, we embarked on a systematic exploration of the influence of two pivotal parameters on the predictive performance of our model: the number of layers within the graph neural network (referred to as K) and the dimension of the embedded representation (represented as S). To comprehensively understand the distinct impact of each parameter, we meticulously conducted individual single-variable control experiments, maintaining all other parameters at a constant level.

We commenced our investigation by scrutinizing the number of layers, denoted as K, within the graph neural network. Through a deliberate manipulation of K ranging from 1 to 5, we meticulously conducted five-fold cross-validation to meticulously assess the resulting prediction outcomes. The specific empirical data and corresponding trend chart are meticulously documented in Table 5 and Figure 3A, respectively. Upon careful analysis, a prominent observation emerged: the model achieved the optimal performance when K was designated as 3. This discerning finding suggests that a judicious choice of the number of layers engenders a harmonious equilibrium between the model complexity and prediction accuracy.

Subsequently, our investigation sought to comprehend the effect of the dimension of the embedded representation, designated as S. Through a meticulous process involving a five-fold cross-validation, we methodically evaluated the model’s predictive efficacy across an array of dimension values: 8, 16, 32, 64, 128, and 256. The empirical data pertinent to this investigation, along with the corresponding trend depiction, are meticulously documented in Table 6 and Figure 3B, respectively. The outcome of this exploration substantiated that the model exhibited its optimal performance at an S value of 64.

### 2.6. Case Studies

This study incorporated comprehensive case studies to substantiate the efficacy of our prediction model, GCLSDA, in forecasting plausible snoRNA–disease associations. The focus was directed towards two specific case studies: colorectal carcinoma and osteosarcoma.

For the colorectal carcinoma [26] case study, a meticulous approach was adopted. We conducted a meticulous selection process based on the association matrix predicted by our model, focusing on prevalent and representative diseases. Initially, we ranked the top 15 snoRNAs according to their association scores as predicted by our model and cross-verified them using the PubMed Identifier (PMID) database. To ensure the credibility of our results, snoRNAs with established associations were deliberately excluded, leaving a pool of candidate snoRNAs for prediction by the GCLSDA model. Subsequently, we sorted these candidate snoRNAs in descending order based on their prediction scores and selected the top 15 for further scrutiny. This rigorous validation process involved an exhaustive examination of the latest literature, encompassing the PubMed dataset, as well as insights gleaned from clinical trials. The outcome of this validation effort, including the number of confirmed snoRNAs, is summarized in Table 7. Remarkably, out of the top 15 candidate snoRNAs, 12 were confirmed and validated. This underscores the robustness of GCLSDA in predicting potential associations between snoRNAs and colorectal carcinoma.

The osteosarcoma [27] case study closely mirrored this methodology. By excluding the snoRNAs with established links, the GCLSDA model was empowered to predict uncharted associations. Similarly, the top 15 candidates derived from these predictions underwent rigorous validation against a backdrop of PubMed literature and clinical trial data. The validation outcomes, encapsulated in Table 7, chronicle the count of validated snoRNAs. Notably, within the elite cohort of top 15 candidates, an impressive 12 snoRNAs were substantiated, further accentuating the predictive potency of GCLSDA in discerning potential snoRNA–osteosarcoma associations.

To further assess the ability of different methods to express biological associations between snoRNAs and diseases, we conducted comparative experiments based on the outcomes of our ablation experiments. Firstly, we trained models using the same dataset as the validation set (with the snoRNA–disease pairs associated in the training set removed) to generate correlation prediction score matrices. Subsequently, we focused on two classic and representative diseases, colorectal carcinoma and osteosarcoma, and extracted the top 15, 20, 25, and 35 snoRNAs with the highest predicted scores for each method. We then verified them in PubMed Identifier database to determine whether publicly available data had verified their correlations. The results are presented in Table 8.

Notably, our method, GCLSDA, not only demonstrated a superior AUC performance in absolute quantitative terms but also proved to be highly adept at identifying biologically significant associations.

It should be noted that, in our experiment, it is found that the prediction results of various methods are small when the top 12 or less than 12 are selected, so 15 is selected as the smallest set. When the top 35 or greater than 35 were selected, the score of the association prediction was too low and considered to be of no research significance.

The case studies conducted underscore the significance of GCLSDA as a valuable tool in pinpointing noteworthy snoRNA candidates warranting deeper investigation within specific disease contexts. The substantial validation rates accentuate the credibility of our approach in expediting the identification of potential snoRNA–disease associations. These findings not only serve as a catalyst for future research endeavors but also hold the potential for clinical applications in the realm of precision medicine.

## 3. Materials and Methods

In this work, we propose GCLSDA, a novel method that combines graph convolution networks and self-supervised learning to predict potential associations between snoRNAs and diseases. GCLSDA takes the snoRNA–disease bipartite graph as input and produces association scores for specific snoRNAs and diseases. Initially, we initialize the embeddings of snoRNAs and diseases randomly. To enhance the data, we introduce random noise to each feature space, facilitating efficient data augmentation. Moreover, we utilize LightGCN to effectively aggregate information from neighboring nodes and refine the uniformity of feature representations through an improved loss function. This approach enables the model to learn more informative and accurate feature representations of snoRNAs and diseases. To address issues of data sparsity and similarity noise, we employ self-supervised learning with cross-layer comparisons in small batches. This strategy ensures the simplicity of single propagation calculations. Finally, we employ the inner product to compute the association scores between specific snoRNAs and diseases. The complete workflow of GCLSDA is illustrated in Figure 4.

### 3.1. Datasets

For known snoRNA–disease associations, we utilized two distinct datasets: MNDR v4.0 [28] and ncRPheno [29]. MNDR v4.0 is a comprehensive and concise resource for RNA disease-associated data, encompassing 3,428,058 RNA disease entries spanning 18 RNA types, 117 types, and 4090 diseases. After eliminating redundant information, we extracted 1441 associations involving 453 snoRNAs and 119 diseases from MNDR v4.0. On the other hand, ncRPheno provides a diverse collection of noncoding RNA and disease associations, encompassing miRNAs, lncRNAs, circRNAs, snoRNAs, and piRNAs. Following the removal of duplicate data, we retrieved 362 snoRNA–disease associations from ncRPheno, consisting of 6 snoRNAs and 119 diseases. In total, our experimental dataset comprised 1538 snoRNA–disease associations, encompassing 456 snoRNAs and 194 diseases.

### 3.2. Problem Description

To achieve the prediction of the relationship between snoRNA and disease, we consider a given set of snoRNAs, denoted by $S=\{s_{1},s_{2},\ldots ,s_{m}\}$, and a set of diseases, denoted by $D=\{d_{1},d_{2},\ldots ,d_{n}\}$. We represent the association between snoRNA and disease using the association matrix $R\in R^{m\times n}$. This matrix captures the known associations between snoRNAs and diseases.

By utilizing the snoRNA–disease association matrix, we construct a bipartite graph denoted by G(S,D,E). In this graph, each edge e=(s,d)∈E represents a known association between the snoRNA *s* and the disease *d*. The presence of an edge indicates that a relationship exists between the snoRNA and the disease.

By representing the data in the form of a bipartite graph, we can employ graph-based methods to analyze and predict the relationships between snoRNAs and diseases.

### 3.3. Initialize Embedding

To learn embedding representations of snoRNA and disease, we utilize a bipartite graph as input and perform the feature aggregation of neighboring nodes. At this stage, we initialize the ID embedding matrices for snoRNA and disease. By conducting an embedding search based on the ID of each snoRNA or disease, we map them into vector representations. By initializing the embedding matrices, we establish a 0-order representation for each snoRNA and disease. These initial embeddings serve as a starting point for subsequent propagation layers, where we prepare to embed them further. In mathematical terms, the ID embedding matrix for snoRNAs and disease can be abstracted as:(5)$$\left\{\begin{array}{c} E_{s}=\left[e_{s_{1}},e_{s_{2}},\ldots ,e_{s_{n}}\right]\\ E_{d}=\left[e_{d_{1}},e_{d_{2}},\ldots ,e_{d_{m}}\right] \end{array}\right.$$

The feature matrices for snoRNA and disease, denoted by $E_{s}$ and $E_{d}$, respectively, represent the initial embeddings for snoRNA and disease. The ID embedding vector $e_{si}\in R^{T}$ represents the embedding of the *i*-th snoRNA, while $e_{dj}\in R^{T}$ represents the embedding of the *j*-th disease. Here, *T* denotes the dimensionality of the feature vectors, determining the size of the embeddings. These feature matrices and ID embedding vectors form the foundation for further embedding propagation and the subsequent analysis of snoRNA and disease relationships.

### 3.4. Graph Convolutional Network for GCLSDA

Once the graph is constructed, preprocessing and feature extraction steps are performed to extract relevant features from the data. Graph convolutional networks (GCNs) [30] aim to learn node representations by smoothing features across the graph. This is achieved through iterative convolutions on the graph, where the features of neighboring nodes are aggregated to create new representations of target nodes. In GCN, feature transformation, domain aggregation, and nonlinear activation are common operations. However, studies [28,31] have shown that the two common designs of feature transformation and nonlinear activation have a limited impact on the effectiveness of collaborative filtering and can increase the complexity of model training. Therefore, in our model, we solely focus on neighborhood aggregation. We utilize a simplified GCN-based embedding propagation layer that employs a weighted-sum aggregator to capture key collaborative filtering signals based on the graph structure. This approach optimizes the embedding representation of snoRNA and disease. Specifically, LightGCN [14] aggregates neighbor nodes at each layer, resulting in the k-layer propagated embeddings for snoRNA *s* and disease *d*, which can be expressed using the following formula:(6)$$\left\{\begin{array}{c} e_{s}^{\left(k\right)}=\sum_{d\in N_{s}}\frac{1}{\sqrt{\left|N_{s}\right|\left|N_{d}\right|}}e_{d}^{\left(k-1\right)}\\ e_{d}^{\left(k\right)}=\sum_{s\in N_{d}}\frac{1}{\sqrt{\left|N_{s}\right|\left|N_{d}\right|}}e_{s}^{\left(k-1\right)} \end{array}\right.$$
where $e_{s}^{\left(k\right)}$ and $e_{d}^{\left(k\right)}$ represent the embeddings of snoRNA *s* and disease *d*, respectively, at layer *k*. The sets $N_{s}$ and $N_{d}$ represent the diseases directly interacting with snoRNAs and the snoRNAs directly interacting with disease *d*, respectively. The term $\frac{1}{\sqrt{\left|N_{s}\right|\left|N_{d}\right|}}$ serves as the normalization coefficient or discount coefficient. It accounts for the decay of information as the propagation path length increases. After K-layer graph convolutions, the embedded representations of each layer are weighted and summed to obtain the final embedded representations of snoRNA *s* and disease *d*:(7)$$\left\{\begin{array}{c} e_{s}=\sum_{k=0}^{K}\alpha_{k}e_{s}^{\left(k\right)}\\ e_{d}=\sum_{k=0}^{K}\alpha_{k}e_{d}^{\left(k\right)} \end{array}\right.$$

In Equation (Equation 7), $\alpha_{k}$ represents the importance of the *k*-th layer embedding in the final embedding representation. It can be manually adjusted or automatically optimized as a parameter. In this experiment, we set $\alpha_{k}$ to $\frac{1}{K+1}$ to ensure good performance.

### 3.5. Contrastive Learning

Traditional unsupervised learning methods, particularly those rooted in graph convolutional neural networks, encounter certain constraints when applied to the prediction of associations in recommender systems [32]. This is particularly evident in the context of the long-tail problem [33,34]. In this scenario, conventional models tend to exhibit a bias towards nodes with higher degrees, a tendency that promotes more effective representation learning for such nodes. However, nodes with lower degrees, often referred to as long-tail nodes, present a more intricate challenge for effective representation learning. Additionally, the conventional practice of aggregating neighborhood nodes exacerbates the impact of noise present in interaction data, potentially compromising the model’s robustness.

When addressing the prediction of snoRNA–disease associations, the prevailing supervised learning signals continue to grapple with issues of data sparsity, impeding the acquisition of refined representations. To mitigate this, we embraced self-supervised contrastive learning [35,36] as our preferred training approach. Contrastive learning stands as an alternative methodology that circumvents the limitations encountered in conventional self-supervised learning paradigms. By doing so, it furnishes a comprehensive framework to enhance both the efficacy and robustness of the model concerning the prediction task.

In the scope of our experiment, in pursuit of fostering greater distinction between diverse node representations and ensuring the discernibility of individual nodes, we adopted a straightforward strategy. Specifically, we introduce random noise directly into the representations to induce a higher level of dissimilarity. For a given node *i* and its associated representation $e_{i}$ within a d-dimensional embedding space, this approach of representation-level augmentation is formulated as follows:(8)$$\left\{\begin{array}{c} e_{i}^{\prime }=e_{i}+\Delta_{i}^{\prime }\\ e_{i}^{\prime \prime }=e_{i}+\Delta_{i}^{\prime \prime } \end{array}\right.$$

The noise vectors $\Delta i^{\prime }$ and $\Delta i^{\prime \prime }$ are randomly generated with ${∥\Delta ∥}_{2}=\epsilon$, where ϵ is a small constant. This noise augmentation strategy aims to promote diversity among node representations by introducing small perturbations into their original embeddings. By doing so, we encourage distinctiveness and discourage similarity between different nodes in the embedding space. To ensure that Δ is numerically equivalent to a point on a hypersphere of radius ϵ, the following constraint is imposed:(9)$$\Delta =Y⊙\mathrm{sign}\left(e_{i}\right),\;Y\in R^{d}∼U\left(0,1\right)$$

In this equation, *Y* is a vector of the same dimensionality *d* as the embedding vector $e_{i}$. Each element of *Y* is independently and uniformly sampled from the interval [0, 1] using the uniform distribution U(0,1). The ⊙ symbol represents the element-wise multiplication operation between *Y* and the sign function applied element-wise to the original embedding vector $e_{i}$.

The sign function, denoted as $\mathrm{sign}(e_{i})$, computes the sign (+1, 0, or −1) of each element in $e_{i}$. By element-wise multiplying *Y* with the sign of $e_{i}$, the resulting noise vector Δ inherits the signs of $e_{i}$. This constraint ensures that Δ lies within the same hyperoctant (super octant) as the original embedding vector $e_{i}$. By constraining Δ in this manner, excessive bias introduced by the noise is prevented, and the augmentation of $e_{i}$ remains informative.

From a geometric standpoint, adding the scaled noise vector to $e_{i}$ corresponds to rotating it by two small angles. Each rotation deviates from $e_{i}$ and generates an augmented representation. These augmented representations preserve most of the information from the original representation while introducing some differences. Furthermore, to fully exploit the expressive power of the embedding space, we aim to spread the learned representations across the entire space. This property is demonstrated by the uniform distribution. Therefore, we choose to generate noise from a uniform distribution. Although it is technically challenging to make the learned distribution approximately uniform, generating noise from a uniform distribution consistently provides hints for augmentation in a statistical manner.

In the contrastive learning framework, our objective is to minimize the feature representation augmented by the same embedding and maximize the feature representation of the different embedding with its added noise. In this experiment, we adopted the classical contrastive learning approach by utilizing the same perturbed representation for both tasks. Cross-layer contrast is employed to simplify the learning process. To facilitate contrastive learning, we incorporate the InfoNCE loss as an auxiliary task. The InfoNCE loss is defined as follows:(10)$$L_{cl}=\sum_{i\in \beta }\log\frac{\exp\left({z_{i}^{\prime }}^{T}\right)z_{i}^{l}/\tau }{\sum_{i\in \beta }\exp\left({z_{i}^{\prime }}^{T}\right)z_{i}^{l}/\tau }$$
where *l* denotes the layer compared to the final layer. In the training batch, *i* and *j* represent the samples of snoRNA and disease, respectively. $z_{i}^{\prime }$ and $z_{i}^{l}$ represent the feature representations of snoRNA after adding random noise, while $z_{j}^{l}$ represents the feature representation of disease. The constant τ>0 is the temperature parameter that controls the strength of the sample penalty. The InfoNCE loss encourages consistency between $z_{i}^{\prime }$ and $z_{i}^{l}$, which are positive samples of each other, while minimizing the consistency between $z_{i}^{\prime }$ and $z_{j}^{l}$, which are negative samples of each other. Optimizing the InfoNCE loss effectively minimizes the mutual information between the representations.

### 3.6. Prediction

Finally, we utilize the embedded representation of snoRNA $s_{i}$ and the embedded representation of disease $d_{j}$ to obtain the final prediction score using the inner product operation:(11)$${\hat{y}}_{s_{i}d_{j}}=e_{s_{i}}^{T}e_{d_{j}}$$

This equation computes the inner product between the embeddings $e_{s_{i}}$ and $e_{d_{j}}$, resulting in the predicted score for the association between snoRNA $s_{i}$ and disease $d_{j}$. The resulting prediction score represents the likelihood or strength of the association between the snoRNA and disease. Higher scores indicate a higher probability of association. By calculating the inner product between the snoRNA and disease embeddings, we capture the compatibility and similarity between their respective representations, enabling us to make predictions about their potential association.

### 3.7. Model Optimization

During training, we employ the pairwise Bayesian personalized ranking (BPR) loss [37] as the loss function. BPR is a sorting algorithm based on matrix factorization that aims to maximize the gap between the scores of positive and negative samples. The BPR loss is calculated as follows:(12)$$L_{BPR}=\sum_{(s,d_{+},d_{-})\in B}-\log({\hat{y}}_{sd_{+}}-{\hat{y}}_{sd_{-}})$$

Among them, *B* represents the training set consisting of triplets $\left(s,d_{+},d_{-}\right)$, where *s* is a snoRNA, $d_{+}$ denotes a positively associated disease, and $d_{-}$ represents a randomly sampled negatively associated disease. The objective of the BPR loss is to ensure that the predicted score ${\hat{y}}_{sd+}$ for the positive association is greater than or equal to the predicted score ${\hat{y}}_{sd-}$ for the negative association.

To incorporate the contrastive learning objective, we combine the BPR loss with the contrastive learning loss using a regularization parameter λ. The overall loss function is given by:(13)$$L=L_{BPR}+\lambda L_{cl}$$

This expression combines the BPR loss and the contrastive learning loss, allowing the model to optimize both the ranking performance and the feature representation learning through self-supervised contrastive learning.

## 4. Conclusions

In conclusion, our self-supervised framework, GCLSDA, represents a significant advancement in predicting snoRNA–disease associations. By harnessing graph convolutional neural networks and incorporating contrast learning, GCLSDA adeptly addresses challenges related to data sparsity and noise, resulting in enhanced prediction precision and model robustness. The comprehensive evaluation of GCLSDA reaffirms its superiority over existing methodologies, highlighting its potential as a valuable and reliable resource for unraveling the functional underpinnings of snoRNAs in disease contexts.

The promising performance of GCLSDA holds profound implications for disease understanding, drug exploration, and therapeutic strategies. Through precise predictions of potential snoRNA–disease associations, GCLSDA opens new avenues for deciphering the intricate pathogenic mechanisms of complex human ailments. This knowledge facilitates the development of refined, individualized diagnostic tools and therapeutic interventions, ultimately improving patient outcomes. As we continue to refine and expand computational tools like GCLSDA, we anticipate transformative impacts on precision medicine. Exploring higher-order associations within the bipartite graph structure and integrating diverse data sources, such as gene expression data, bodes well for future exploration. Further enhancements to snoRNA and disease embeddings through self-supervised learning will continue to amplify GCLSDA’s predictive potency and real-world applicability.

However, it is worth noting that our current method has limitations, particularly in terms of database scale. While our approach performs efficiently on relatively small databases, its scalability to larger datasets may present challenges in terms of computational resources and optimization. Addressing these limitations will be a key focus of our future work, as we strive to extend the applicability of GCLSDA to more extensive datasets and further advance the field of computational snoRNA–disease association prediction.

## Figures and Tables

**Figure 2 ijms-24-14429-f002:**
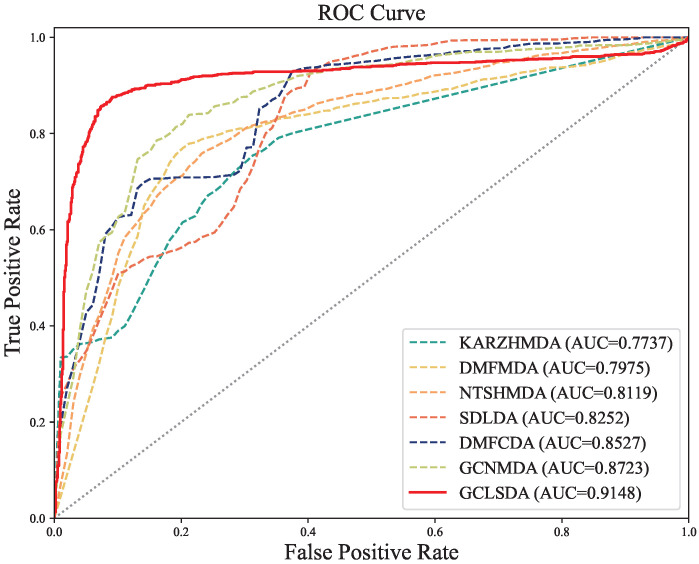
ROC curves of GCLSDA compared with six benchmarked methods.

**Figure 3 ijms-24-14429-f003:**
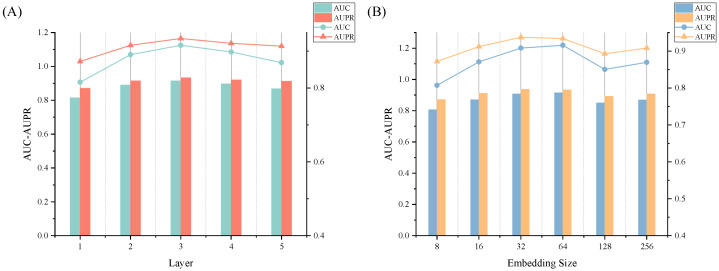
Effects of depth K and embedding size S on the performance of GCLSDA. (**A**) The effect of layer depth K. (**B**) The effect of embedding size S.

**Figure 4 ijms-24-14429-f004:**
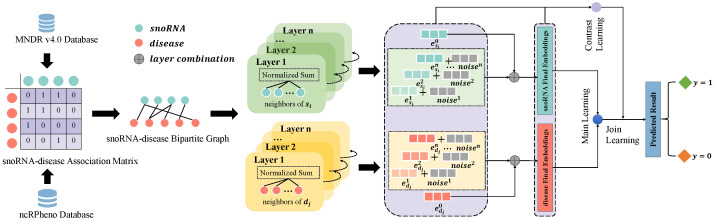
Workflow of GCLSDA. The process initiates with the collection of established associations between viruses and drugs, sourced from the MNDR v4.0 and ncRPheno databases. Subsequently, a bipartite graph is meticulously constructed, delineating the interrelationships between snoRNAs and diseases, based on these gathered associations. GCLSDA operates through two pivotal tasks: the main task and the contrastive task. The main task involves the introduction of the bipartite graph into the LightGCN framework. Within this context, the aim is to acquire meaningful representations of the distinct snoRNA and disease nodes. These acquired representations are then harnessed in conjunction with the inner product to prognosticate associations between snoRNAs and diseases. Owing to the sparse nature of the supervised signals within the dataset, coupled with the presence of similarity noise, the efficacy of the learned snoRNA and disease representations is further augmented. This is accomplished by incorporating cross-layer comparisons within limited batches, thereby fostering self-supervised learning. Ultimately, the main and contrastive tasks are harmoniously integrated, culminating in enhanced predictive capabilities pertaining to the associations between snoRNAs and diseases.

**Table 1 ijms-24-14429-t001:** Five-fold cross-validation prediction results of GCLSDA.

K	AUC	AUPR
0	0.9151	0.9471
1	0.9224	0.9508
2	0.9322	0.9267
3	0.9166	0.9197
4	0.8928	0.9262
AVG	0.9148	0.9354

**Table 2 ijms-24-14429-t002:** Ten-fold cross-validation prediction results of GCLSDA.

K	AUC	AUPR
0	0.9296	0.9540
1	0.9667	0.9756
2	0.9192	0.8946
3	0.9365	0.9264
4	0.9266	0.9507
5	0.9099	0.9398
6	0.9568	0.9581
7	0.9129	0.9379
8	0.9395	0.9489
9	0.8699	0.8997
AVG	0.9260	0.9383

**Table 3 ijms-24-14429-t003:** Performance comparison of GCLSDA and six benchmarked methods.

Methods	AUC	AUPR
KARZHMDA	0.7737	0.8066
DMFMDA	0.7975	0.8183
NTSHMDA	0.8252	0.8228
SDLDA	0.8119	0.8201
DMFCDA	0.8527	0.8401
GCNMDA	0.8723	0.8801
GCLSDA	0.9148	0.9354

**Table 4 ijms-24-14429-t004:** Ablation study of GCLSDA.

Methods	AUC	AUPR
LightGCN	0.8040	0.8497
SGL(ND)	0.8635	0.9115
SGL(ED)	0.8540	0.9047
GCLSDA	0.9148	0.9354

**Table 5 ijms-24-14429-t005:** Prediction results of GCLSDA with different depth K.

Layer	AUC	AUPR
1	0.8157	0.8720
2	0.8905	0.9158
3	0.9159	0.9341
4	0.8976	0.9210
5	0.8689	0.9135

**Table 6 ijms-24-14429-t006:** Prediction results of GCLSDA with different embedding size S.

EM	AUC	AUPR
8	0.8073	0.8720
16	0.8709	0.9123
32	0.9081	0.9374
64	0.9159	0.9341
128	0.8505	0.8928
256	0.8695	0.9083

**Table 7 ijms-24-14429-t007:** The top 15 snoRNAs related to colorectal carcinoma and osteosarcoma resistance predicted by GCLSDA.

SnoRNAs (Colorectal Carcinoma)	Evidence (PMID)	SnoRNAs (Osteosarcoma)	Evidence (PMID)
SNHG12	32606771	TERC	27207662
SNHG20	35915794	SNHG16	33823834
SNHG5	35166647	SNHG7	35422168
SNHG4	34717631	SNHG15	35261804
RNU1-1	36973786	SNHG20	30120876
SNORD50	Unconfirmed	SNORD50	Unconfirmed
SNORD68	Unconfirmed	SNHG12	35422168
SNHG14	31799655	SNHG4	36004691
SNORD115-21	36973786	SNHG8	36388161
SNORD42B	Unconfirmed	SNHG14	36599973
SNORD115-33	36973786	SNHG3	33292213
SNORD115-8	36973786	SNORD24	Unconfirmed
SNORD116-15	36973786	SNHG1	35130629
SNORD93	36973786	SNORD3A	32599901
SNORD115-40	36973786	SNORD43	Unconfirmed

**Table 8 ijms-24-14429-t008:** Confirming the biological relevance of snoRNAs associated with colorectal carcinoma and osteosarcoma through ablation study.

Methods	Colorectal Carcinoma				Osteosarcoma			
	15	20	25	35	15	20	25	35
LightGCN	6	7	7	7	6	8	12	15
SGL(ND)	9	12	15	16	7	7	7	14
SGL(ED)	12	16	18	25	7	8	9	10
GCLSDA	12	16	21	26	12	14	15	18

## Data Availability

The code and datasets of GCLSDA are available at https://github.com/ZhangLiangliangCSU/GCLSDA, (accessed on 14 August 2023).

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
