# Peer review of "Graph Convolutional Network and Contrastive Learning Small Nucleolar RNA (snoRNA) Disease Associations (GCLSDA): Predicting snoRNA–Disease Associations via Graph Convolutional Network and Contrastive Learning"

_ijms, 2023, doi:10.3390/ijms241914429_

Round 1
Reviewer 1 Report
This paper addresses an important and interesting problem: developing predictive models for associations between small nucleolar RNAs (snoRNAs) and disease. The authors propose an approach based on a light graph convolutional network (LightGCN). I should point out that the author had in 2021 published an IEEE conference proceeding describing a Graph Convolutional Network (GCN) based framework. The authors should cite that work. It appears as though the present manuscript builds on that work by employing self-supervision to address the challenge of sparse data, as well as utilizing a LightGCN architecture rather than conventional GCN. I would suggest that the authors add some reference to the prior work as part of the motivation for the manuscript.
Most importantly, on revision, I suggest a thorough review of the English in the manuscript. The wording is very overly complicated and long-winded, or even totally inapposite, in many instances. For example, in the Abstract alone, there are such phrases as “[a]ddressing the intricacies of sparsity and over-smoothing” (no need for “the intricacies”), “ingeniously unite” (drop the adverb “ingeniously”), “consistently unveils” (“consistently shows”?), “meticulously devised” (whole phrase can be deleted) and “strategic amalgamation” (“combination”?). I suspect the potential use of machine translation or AI tools to draft the manuscript – I do not think that this is a problem, and I sometimes utilize LLMs myself to assist in drafting. However, careful review and revision of the language is still necessary. I do note that the authors’ previous 2021 work did not nearly have as much of this issue with overly complicated and flowery language.
Regarding substantive issues, I do not see a reference in the manuscript as to where to see the code that the authors developed, or even what software packages or hardware were used. Making the code publicly available is in the spirit of this being an open access journal and it is also essential for ensuring the reproducibility of results. The experimental dataset likely should also be made available, or at least the identifiers of the data used in the RNA-disease databases that show specifically which data from the databases the authors utilized (if not the raw, underlying data itself).
Another point is that the 2021 conference paper appears to have used the same experimental data set. Has any additional data been generated or made available since then? If so, then this could be used as out-of-domain test data (which makes a stronger point than cross-validation), or it can be used to augment training. If it is available, I strongly suggest out-of-domain testing to further validate the results, since it is possible to do overfitting even in cross-validation (for example by optimizing hyperparameters on the same data set, or because there are associations across folds of the data set if they cannot be truly independent, etc.).
I do not know if the case studies are sufficient to mitigate this concern. I note that they are from 2017 and 2010. Can the authors clarify if data from the case studies is included in the training data set? If not, how do they know, i.e., did they verify with the databases? It would be more persuasive if the authors could rely on case studies after their training data were obtained, again, to mitigate the issue of overfitting. Moreover, the authors should show how other methods perform in the case study. Is the LightGCN/contrastive learning method better able to find biologically significant associations, or is it relatively similar (even if absolute quantitative performance may be better on the AUC metric)?
The wording is very overly complicated and long-winded, or even totally inapposite, in many instances. For example, in the Abstract alone, there are such phrases as “[a]ddressing the intricacies of sparsity and over-smoothing” (no need for “the intricacies”), “ingeniously unite” (drop the adverb “ingeniously”), “consistently unveils” (“consistently shows”?), “meticulously devised” (whole phrase can be deleted) and “strategic amalgamation” (“combination”?). I suspect the potential use of machine translation or AI tools to draft the manuscript – I do not think that this is a problem, and I sometimes utilize LLMs myself to assist in drafting. However, careful review and revision of the language is still necessary. I do note that the authors’ previous 2021 work did not nearly have as much of this issue with overly complicated and flowery language. I want to emphasize that while my examples are from the Abstract, this continues throughout the manuscript.
Reviewer 2 Report
In the article the authors describe a framework termed GCLSDA, based on light graph convolutional network and contrastive learning to predict snoRNA-disease associations. The authors have shown that GCLSDA performance was superior when compared to other benchmark methods previously published. Also, the two case studies investigated showed remarkable efficiency in identifying disease association of snoRNAs tested, based on literature data. Please see below a few comments to improve the reading experience of the manuscript.
· Line 6: What does “GCLSDA” stand for? Please define once when first used.
· Line 98: Briefly describe Light Graph Convolutional Network (LightGCN) in main text.
· Figure 1: Please increase the font size of “Disease Final Embeddings” and “snoRNA Final Embeddings” as it is difficult to read.
· Line 354: It is supposed to be Figure 3 instead of Figure 4.
· Please define acronyms in all figure legends.
· In section 3.5: Though the ablation experiment methodology was described in detail in the main text, the results are not discussed. It would be useful to add a brief discussion of the results for performance comparison (instead of only referring to Table 4).
· Line 422: On what basis was ranking performed? Please describe. If a few snoRNAs are picked from the bottom ranks, do you expect to find disease association? Based on the case study, what should be the threshold to obtain meaningful disease association?
· What are some of the challenges associated with this methodology? How can this be improved or what is the future?
· Throughout the text “Light Graph Convolutional Network (LightGCN)” was defined multiple times (Line 98, 104, and 369). You need to define it only once when first used.
· Line 306: Define TPR and FPR as True positive rate and False positive rate respectively.
Round 2
Reviewer 1 Report
Thank you for addressing the issues raised in the first round of review. I would still consider reviewing and simplifying some of the languages, but as far as I can tell there are no obvious inaccuracies.